**Data Availability Statement:** all nessary dats avilable in the manuscript

**Funding:** The research supported by budget by Amhara Agricultural Research Institute (ARARI)

# Effects of dry bio-slurry and nitrogen fertilizer on potato and wheat yields under rotation cropping system

**Zelalem Addis**[1]*, **Tadele Amare**[1], **Bitewlgn Kerebih**[1], **Anteneh Abewa**[1], **Tesfaye Feyisa**[2], **Abrham Awoke**[1], **Abere Tenagne**[1]

1 Adet Agricultural Research Center, Bahir dar, Ethiopia, 2 Amhara Agricultural Research Institute, Bahir dar, Ethiopia

* zelalemaddis660@gmail.com

## Abstract

Integrated nutrient management and crop rotation are important farming practices, which enhance the nutrient use efficiency of crops and reduce the incidence of diseases and insect pests. The study was carried out to address the gap in using integrated nutrient management in crop rotation systems for soil qualities and crop yield improvement. That was done by adjusting the balance ratio of dry bio-slurry and nitrogen fertilizers. The experiment was containing ten levels; Control (0,0), recommended nitrogen, 50% dry-bio slurry, 100% dry-bio slurry, 75% dry-bio slurry, 75% dry-bio slurry+25% recommended nitrogen, 50% dry-bio slurry+50% recommended nitrogen, 25% dry-bio slurry+75% recommended nitrogen, 100% dry-bio slurry + 25% recommended nitrogen and 100% dry-bio slurry + 50% recommended nitrogen that was laid out in randomized complete block design with three replications for three years. The data on soil properties and yield components of potatoes and wheat were collected and analyzed using statistical analysis system software 9.4. An application of dry bio-slurry with nitrogen fertilizer was significantly affected both crop yield and soil properties in the rotation system. The application of 25% dry bio-slurry with 75% recommended nitrogen gave the highest tuber yield of potato (27.6 tha⁻¹) as compared to control. Similarly, using 100% and 75% sole dry bio-slurry resulted in the highest grain yield (3.85 tha⁻¹) and above-ground biomass (9.59 tha⁻¹) of wheat. The combination of 25% bio-slurry with 75% recommended nitrogen scored the highest net benefit (2889.2 US$) with an acceptable marginal return (4463.3%) via by improving crops yield in the system. So, an application of 25% dry bio-slurry with 75% recommended nitrogen could be promoted for yield-soil improvement in the study area and similar agroecology.

## 1. Introduction

A well-planned cropping system can enhance nutrient use efficiency, reduce the need for inorganic fertilizers and lessen the impact of pests and diseases [1, 2]. Potato and wheat production under a crop rotation system is commonly practiced by farmers in the Amhara region. The

and the funder hand no role in data collection and designs and none of the authors had not receive salary from the funder.

**Competing interests:** The authors have declared that no competing interests exist.

system comprises a cradle-to-farm analysis for a 3-years rotation cycle. It allows for mitigating the impact of certain pests and diseases, which significantly affect the yield of the main crop potato. In addition to this, it enables to take full advantage of residual fertility that comes from potato leaves' entire decomposition in the soil. This rotation system was chosen by many producers and it is considered as the most efficient system from an economic, quality and soil health point of view. Potato (*Solanum tuberosum* L.) is the third most important food crop after wheat and rice in the world [3]. In Ethiopia, the area coverage under potato cultivation reaches about 73,677.64 ha and its production was estimated at around.1,044,436.359 tons [4]. The productivity of potato in Ethiopia has reached about 13.9 t ha$^{-1}$ [5] which, is relatively low compared to other African countries [6]. Bread wheat (*Triticum aestivum* L.) is also one of the major cereal crops grown in the highlands of Ethiopia and it makes the country regarded as the largest wheat producer in Sub-Saharan Africa [7].

Out of the total grain crop area coverage, wheat ranked 4th after TEF (*Eragrostis tef*), maize (*Zea mays*) and sorghum (*Sorghum bicolor*) while it is the third in total production after maize and tef [5]. Despite the long history of wheat cultivation and its importance to Ethiopian agriculture, its average yield is also still very low, not exceeding 2.4 t ha$^{-1}$ [5]. Which is below the world's average yield of wheat 3.4 t ha$^{-1}$ (6). This phenomenon is common in the Amhara region. Low level of potato-wheat productivity is mainly due to soil fertility degradation, improper fertilization, poor pest management practices, use of the low-quality seed and soil nutrient depletion [8]. Enhancing soil fertility is the first precondition for a practical crop production system; that can be achieved through the application of organic manure (like bio-slurry) and inorganic fertilizers integration for sustainable crop productivity [9, 10]. Different studies also indicated that the judicious application of both organic and inorganic fertilizers is a key solution for crop productivity and soil fertility enhancement. The study conducted by [11] revealed that integrated application of both organic and inorganic source fertilizers significantly improved yield and soil fertility status under the maize-potato cropping system. Another study which was done by [12] indicated that the supplement of tricho compost with chemical fertilizer and vermicompost with chemical fertilizer significantly increased the yield of potato, mung bean and T.aman rice by the scoring of 6.3–33.7%,8.3–33.8% and 2.9–18.3% respectively over sole application treatment. Similarly, the study conducted by [45] also observed that poultry manure bio-slurry, poultry manure, cow dung bio-slurry, and cow dung gave 11.7, 8.9, 5.4, and 3.1%. This makes, a respective increment in total system productivity over sole chemical fertilizer (46% N as urea, 20% P as triple super phosphate, 50% K as muriate of potash and 18% S as gypsum). Bio-slurry obtained after extraction of the energy content of animal manure is an excellent fertilizer, it is rich in major nutrients (nitrogen, phosphorous and potassium) and organic matter which is important for soil fertility and yield of crops [13]. It also improves the physical and biological quality of soil beyond its role in the provision of plant nutrients. In addition, an application of bio-slurry can help in the reduction of dependence on mineral fertilizers [14]. Because nitrogen is the most limiting essential nutrient for the growth and development of crops, both potato and wheat are highly responsive to N fertilization [15]. Furthermore; supplying nitrogen fertilizer plays an important role in the balance between vegetative and reproductive growth of crops [16, 17]. Various studies showed that N fertilizer applications can increase the dry matter protein content of wheat and potato tubers [18]. Moreover, most of the time the available nitrogen concentration in most organic source fertilizers including bio-slurry is high as compared to other major nutrients [13]. Due to this, the application of dry bio-slurry with nitrogen fertilizer through a nitrogen equivalence balancing ratio is very important. Besides this, neither organic manure nor chemical fertilizer alone can meet the nutrient demand of a given crop and also their negative impact in terms of sustainable crop and soil productivity in different cropping systems [19]. For instance, the

continuous addition of chemical fertilizers can cause soil quality deterioration such as increasing soil acidity, loss of organic matter and depletion of nutrients that are not supplied in the fertilizer formulation [20]. On the other hand, organic fertilizers are required in large quantities and their nutrients are released slowly making less efficient on soil properties and crop yields within a short period as compared to inorganic fertilizers [21]. Because of these reasons, the study was conducted to determine the main and residual effects of dry-bio slurry and nitrogen fertilizers on soil and yield of potato and wheat under a crop rotation system at Yilemana Densa District in North Western Amhara Region Ethiopia.

## 2. Material and methods

### 2.1 Description of study area

The study was conducted at Yilemana Densa district on a Farmer's field across three sites for three years (2019–2022) as a potato and wheat rotation system in Amhara Region Ethiopia. Geographically the area lies at 11˚ 21' 18" to 11˚ 21' 22" N and 37˚ 25' 37" to 37˚ 25' 43" E [Fig 1] with a mean altitude of 2304 m above sea level. It receives a mean annual rainfall of 1421 mm with mean minimum and maximum temperatures of 12.29 and 27.56˚C, respectively. The landforms of the area are characterized by undulating to rolling plateaus, scattered moderate hills, dissected side slopes, and river gorges [22]. Based on the District Bureau of Agriculture, the major land use comprises cultivated land (57%), forest and bushes (2%), grazing land (33%), and others (8%). Major crops, grown in the study area are Maize, Tef, wheat, Barley, Potato, and Field pea. Soil types in the area are Nitisols (45%) Vertisols (30%) and Luvisols (25%). This on-farm experiment was conducted on Luvic Nitisols which is the most dominant soil in the study area.

### 2.2 Experimental procedure

The experiment was contained ten treatments which include control (without nitrogen and dry bio-slurry), recommended nitrogen (138 N), dry-bio slurry 50% equivalence (5.3 tha$^{-1}$), dry-bio slurry 100% equivalence (10.6 tha$^{-1}$), dry-bio slurry 75% equivalence (7.95 tha$^{-1}$), dry-bio slurry 75% equivalence (7.95 tha$^{-1}$) + 25% recommended nitrogen (34.5 N), dry-bio slurry 50% equivalence (5.3 tha$^{-1}$) + 50% recommended nitrogen (69 N), dry-bio slurry 25% equivalence (2.65 tha$^{-1}$) + 75% recommended nitrogen (103.5 N), dry-bio slurry 100% equivalence (10.6 tha$^{-1}$)+25% recommended nitrogen (34.5 N) and dry-bio slurry 100% equivalence (10.6 tha$^{-1}$) +50% recommended nitrogen (69 N) that were laid out in Randomized Complete Block Design with three replications. The rates of dry-bio slurry were adjusted based on the potato nitrogen recommended rate 138 kgha$^{-1}$ N [23] equivalency corresponding to its nitrogen content. Urea was used as a source of synthetic 138 N whereas, phosphorous (P$_2$O$_5$) at the rate of 69 kg ha$^{-1}$ was applied in the form of TSP to all plots. The experiment was carried out under rain-fed conditions GUDENIE and TAY varieties were used as test crops for potato and wheat respectively in each rainy season one after the other. The total area of each plot was 4.5 m x 3 m (13.5 m$^2$) having 1 m space between plots and 1.5 m between blocks. Potato was spaced by 0.3 m between plants & 0.75 m between rows and the data were collected from the middle four rows. Whereas; Wheat was planted as a rotating crop in 0.2 m and harvested from middle rows by avoiding four rows as a border.

 **2.2.1 Dry bio-slurry.** It was collected from farmers biogas plants and stored under well-protected shade for better drying and avoiding nutrient loss, especially nitrogen. After drying, we make it fine and break in to small pieces for nice application and soil reaction. After this; During the planting period dry-bio slurry was incorporated into the soil. The representative composite sample was taken from the whole collected dried pit of slurry [Table 1]. For analysis

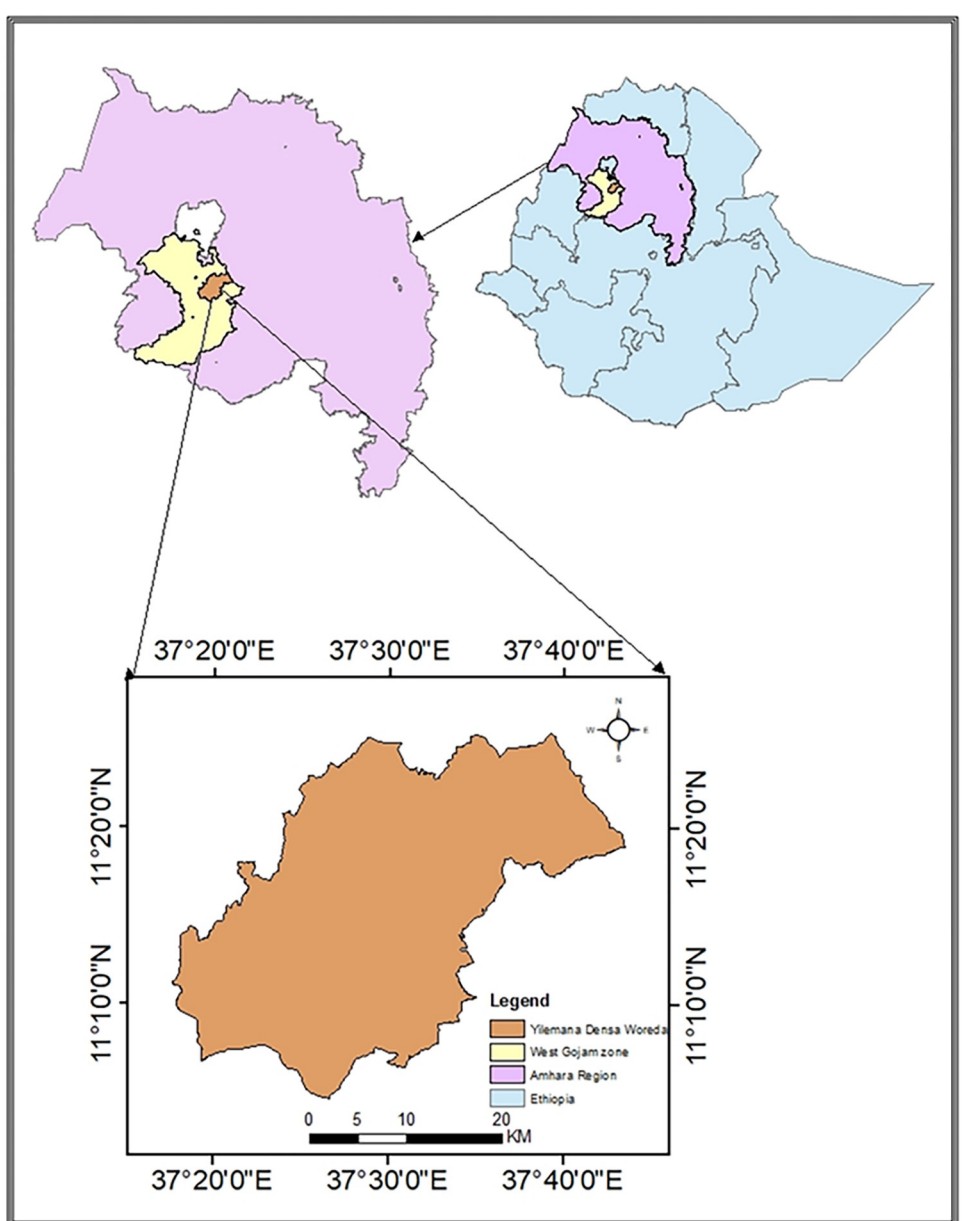

**Fig 1. Study area map.**

of hydrogen concentration (pH), organic carbon (OC%), cation exchange capacity (CEC), total nitrogen (TN%) and available phosphorus (P) by following laboratory manual procedures [24].

**2.2.2 Experimental crops.** **Gudenie** variety is used as a test crop for a potato and it was planted as the main crop or precursor crop of wheat. TAY variety of wheat was also planted in the next cropping season after the potato which was used for the determination of dry bio-slurry residual effect in the potato-wheat rotation system. TSP ($P_2O_5$) was applied during the planting period as basal whereas; Inorganic N from urea was applied in three splits for a potato; One-third third at planting, one-third 30 days after planting, and the remaining one-third at the beginning of the flowering. For wheat recommended NP was applied in all plots

**Table 1. Physico-chemical analysis of dry-bio slurry before incorporation in year1and 2 and its average.**

| Dry bio-slurry | Concentration |
|---|---|
| **Year 1** | |
| Dry matter% | 11.2 |
| OC% | 27.8 |
| TN% | 1.2 |
| Av P (mg kg$^{-1}$) | 112.1 |
| pH (H$_2$O; 1:2.5) | 7.9 |
| C:N ratio | 22.9 |
| CEC (cmol kg$^{-1}$) | 59.8 |
| **Year 2** | |
| Dry matter% | 11.8 |
| OC% | 16.3 |
| TN% | 1.4 |
| Av P (mg kg$^{-1}$) | 91.6 |
| pH (H$_2$O; 1:2.5) | 7.7 |
| C:N ratio | 11.6 |
| CEC (cmol kg$^{-1}$) | 67.4 |
| **Average result of Year1andYear 2** | |
| Dry matter% | 11.5 |
| OC% | 22.1 |
| TN% | 1.3 |
| Av P (mg kg$^{-1}$) | 101.9 |
| pH (H$_2$O; 1:2.5) | 7.8 |
| C: N ratio | 17.3 |
| CEC (cmol kg$^{-1}$) | 63.6 |

OC% = organic carbon percent, TN% = total nitrogen percent, C: N = carbon to nitrogen ratio, CEC = cation exchange capacity, AvP = available phosphorus, and pH = Power of hydrogen concentration.

except control which only received P$_2$O$_5$ to see the residual effect of dry bio-slurry in the next crop and nitrogen was supplied, in two splits.

## 2.3 Data collection, preparation, and analysis

**2.3.1 Soil sampling and analysis of before planting.** Before planting, representative soil samples were collected from 0–20 cm depth in the sites via a random sampling method from 10 spots using an auger. All samples were mixed to form one composite sample. The sample was grounded by using a mortar and passed through a 2 mm sieve for analysis of soil texture, CEC, pH, and available P whereas, a 0.5 mm sieve was used for determining the organic carbon (OC) and total N. While, bulk density was determined by the core sampling method.

**2.3.2 Soil sampling and analysis after harvesting of potato and wheat.** The major chemical properties of soil such as OC, pH, CEC, total N and available P were analyzed following the compiled laboratory manual [24]. Soil pH was measured in water with a ratio of 1:2.5 using a glass electrode pH meter. The soil OC content was determined following the wet digestion method as outlined by Walkley and Black which involves the digestion of soil OC with potassium dichromate (K$_2$Cr$_2$O$_7$) in a sulfuric acid solution. Available phosphorus (AvP) was determined by the Olsen extracting method. The total N content in the soil samples was determined following the Kjeldahl method. CEC was also, determined by extracting the soil samples

with ammonium acetate (1NNH$_4$OAc) followed by repeated washing with ethanol (96%) to remove the excess ammonium ions in the soil solution. Percolating the NH4$^+$ saturated soil with sodium chloride would displace the ammonium ions adsorbed in the soil and the ammonium liberated from the distillation was titrated using 0.1N NaOH.

**2.3.3 Crop data collection.** Plant height for both potato and wheat were measured at the maturity stage by taking five randomly selected plants from the ground to the top apex and averaging for a single reading. Similarly, the spike length of wheat was measured at the maturity stage by taking five randomly selected plants from starting point of the spike grain to the end of the spike grain and averaging for a single reading. The number of plants per hill (stem number per hill) was recorded by counting the stems that were directly sourced from the original seed tuber of five hills of potato at the 70% maturity stage.

The number of tubers per plant was also taken at the maturity stage by counting potato tubers from five randomly selected plants in the middle four rows of experimental plots (7.2 m$^2$) and averaged for a single reading. Similarly, a counted thousand seed weights of wheat were measured using a sensitive balance in each treatment harvestable plot area (11.7 m$^2$). The total tuber yield of potato was measured by using a balance during the harvesting period of time from the middle four rows (7.2 m$^2$) for total weight reading. Grain yield and above-ground biomass of wheat were measured by using hanging scales at maturity stage from the net harvestable plot area (11.7 m$^2$).

## 2.4 Economic analysis

The economic analysis was performed to make rational choices among the applied variables in the production of a potato. The partial budget and marginal rate of return (MRR) were used for evaluating the change in farming methods that affect partially rather than the whole farm practice and also concerned with a planning tool to estimate the profit change within a farm [25]. This was computed by adjusting the yield downward by 10% and multiplying it with the local field price (6 Ethiopian Birr or 0.12 US$ per kg of a potato). Dominance analysis was done by listing treatments in increasing order of cost and that have net benefit less than or equal to treatments with the lower costs that vary is dominated [25].

## 2.5 Statistical analysis

All collected data were subjected to analysis of variance through the general linear model (GLM) a procedure by using the Statistical Analysis System (SAS) software program version 9.4 (SAS Institute, 2014). List significant difference (LSD) at 0.05 probability level was employed to differentiate treatment means t [26].

## 3. Results and discussion

### 3.1 Soil chemical properties before planting and after harvesting of potato

The results of soil chemical properties before and after harvest from all experimental site were displayed in Tables 2–4. Before planting the soil was acidic in reaction with a pH (H$_2$O 1:2.5) value of 5.1 Y1S1 (year one site one) and 5.2 Y2S1 (year two site one), even if, it was within the range of soil pH for potato production [27] (Table 2). The total N, available P, OC, C: N ratio and CEC of the soil Y1S1 and Y2S1 before planting were 0.16&0.16%, 6.9&6.3 mg kg$^{-1}$, 1.4 &1.5%, 11.1&9.3 and 33.9&30.3 cmol (+) kg$^{-1}$, respectively (Table 2). The total N content of the soil was within the range of medium according to [27] who classified the range of total N < 0.1, 0.1–0.15, 0.15–0.25 and > 0.25% as very low, low, medium and high, respectively. [28] also, classified the range of soil available P content was like this < 5 as very low, 5–15 as

**Table 2. Initial soil properties of the experimental sites.**

| Texture SCL | Years | Initial soil properties across sites and years | | | | | | |
|---|---|---|---|---|---|---|---|---|
| Sandy = 56% Clay = 32% Loam = 12% | | BD | pH | TN% | Av P (ppm) | CEC (cmol kg$^{-1}$) | C: N | OC% |
| | 2019–20 (Y1S1 | 1.22 | 5.1 | 0.16 | 6.9 | 33.9 | 8.8 | 1.4 |
| | 2020–21 (Y2S1) | 1.33 | 5.2 | 0.16 | 6.3 | 30.3 | 9.3 | 1.5 |
| | 2020–21 (Y2S2) | 1.26 | 5.5 | 0.17 | 6.8 | 33.9 | 11.8 | 2.0 |

NB: SCL = sandy clay loam; BD = bulk density; Y1S1 = year one site one, Y2S1 = year two site one and Y2S2 = year two site two.

low, 15–25 as medium and > 25 mg kg$^{-1}$ as high. Hence, the available P of the soil before planting lies under the low range. According to [29] the soil OC content ranges were rated 1–2, 2–4, and 4–6% as low, medium and high, respectively. Similarly, cation exchange capacity (CEC) was also rated with the ranges of 5–15, 15–25 and 25–40 cmol kg$^{-1}$ as low, medium and high, respectively [29].

Based on these ratings OC (1.4, 1.5 &2.0%) and CEC (33.9, 30.3&30.9 cmol kg$^{-1}$) before planting of the experimental fields (Y1S1, Y2S1 and Y2S2) were in the low and high ranges, respectively. Generally, the nutrient contents of the study site Y2S1 are not good in terms of the availability of major plant nutrients besides its nice CEC. On the other hand, on Y2S2 pre —planting values of pH, TN, AvP, OC, C: N and CEC were 5.5, 0.17%, 6.8 ppm, 2.4%, 14.1 and 33.9 respectively. Based on [27] rating the pH value was under moderate while total nitrogen was medium. Similarly, according to [28] available P was low while OC and CEC were medium and high respectively [29]. Y2S2 rather than the two sites (Y1S1 and Y2S1) had a good soil fertility status based on its soil chemical properties. However, the acidity of soil Y1S1 and Y2S2 may be causing the sorption of available P. Thus, the application of OM-like bio-slurry is very essential to neutralize soil solution for the availability of nutrients. On the other hand, after harvest, all soil chemical properties except soil pH (on Y2S2) were not affected by the application of dry bio-slurry with N equivalence rates (Table 3).

The non-significant effects of the applied treatments on the rest soil properties might be the slow release of nutrients from the dry bio-slurry that was applied during the experimentation period to the soil solution. However, applied dry bio-slurry and N rates had a significant ($P < 0.05$) effect on soil pH at site two in 2020–21 as compared to the control (Table 2). Numerically the highest soil pH was obtained by the application of 100% dry bio-slurry and 100% dry bio-slurry +25% RN as compared to the control (Table 3). The increment of soil pH in treated plots may be related to the releasing of basic cations from dry bio-slurry into soil solution that makes for the substitution of acid cations. These results agreed with the investigation of [30] which reported that the combined application of compost with inorganic NPSB on maize increased the soil pH content after harvest as compared to the control.

### 3.2 Residual effect of dry bio-slurry on soil chemical properties after harvesting of wheat

The residual effect of dry bio-slurry on selected soil chemical properties was significantly explained at (p < 0.05) across sites and years in Table 4. Related to this an addition of 100% (10.6tha$^{-1}$) dry bio-slurry with 50% RN (69 kg N) was given the highest TN% (0.205 and 0.223) on Y1S1 and Y2S2 respectively than control. This might be due to the gradual release of dry bios-slurry nitrogen into soil solution beyond its chelating capacity. The finding lined with [31] found that an application of dry bio-slurry with inorganic fertilizers gives the maximum value of total nitrogen than control. Similarly, the study conducted by [32] also indicated that

**Table 3. Main effects of dry bio slurry with nitrogen on soil chemical properties after harvest of potato.**

| 2019–20 (Y1S1) | | | | | | |
|---|---|---|---|---|---|---|
| Treatment | pH | TN% | Av P (ppm) | CEC (cmol kg$^{-1}$) | C: N | OC% |
| Control (0,0) | 5.13 | 0.12 | 9.3 | 29.1 | 11.6 | 1.4 |
| RN (138 N kgha$^{-1}$) | 5.11 | 0.12 | 7.8 | 29.9 | 11.61 | 1.3 |
| 50% DBS (5.3 tha$^{-1}$) | 5.19 | 0.13 | 8.8 | 29.6 | 11.59 | 1.4 |
| 100% DBS (10.6 tha$^{-1}$) | 5.18 | 0.13 | 9.6 | 29.7 | 11.58 | 1.5 |
| 75% DBS (7.95 tha$^{-1}$) | 5.19 | 0.13 | 10.2 | 29.1 | 11.62 | 1.5 |
| 75% DBS+25% N (7.95 tha$^{-1}$+34.5 N kgha$^{-1}$) | 5.22 | 0.16 | 10.1 | 28.3 | 11.57 | 1.9 |
| 50% DBS+50% N (5.3 tha$^{-1}$+69 N kgha$^{-1}$) | 5.10 | 0.15 | 9.1 | 28.7 | 11.59 | 1.7 |
| 25% DBS+75% N (2.65 tha$^{-1}$+103.5 N kgha$^{-1}$) | 5.18 | 0.15 | 7.5 | 29.2 | 11.61 | 1.7 |
| 100% DBS+25% N (10.6 tha$^{-1}$+34.5 N kgha$^{-1}$) | 5.12 | 0.13 | 12.3 | 28.3 | 11.62 | 1.5 |
| 100% DBS+50% N (10.6 tha$^{-1}$+69 N kgha$^{-1}$) | 5.15 | 0.13 | 11.2 | 27.6 | 11.60 | 1.5 |
| **LSD** | **NS** | **NS** | **NS** | **NS** | **NS** | **NS** |
| **CV%** | **1.2** | **14.3** | **22.43** | **3.3** | **0.21** | **14.19** |
| 2020–21 (Y2S1) | | | | | | |
| Treatment | pH | TN% | Av P (ppm) | CEC (cmol kg$^{-1}$) | C: N | OC% |
| Control (0,0) | 5.2 | 0.15 | 4.4 | 27.3 | 11.59 | 1.7 |
| RN (138 N kgha$^{-1}$) | 5.3 | 0.16 | 4.3 | 27.7 | 11.59 | 1.8 |
| 50% DBS (5.3 tha$^{-1}$) | 5.3 | 0.14 | 6.5 | 30.8 | 11.61 | 1.7 |
| 100% DBS (10.6 tha$^{-1}$) | 5.4 | 0.15 | 6.2 | 27.4 | 11.60 | 1.8 |
| 75% DBS (7.95 tha$^{-1}$) | 5.4 | 0.15 | 5.8 | 28.6 | 11.58 | 1.7 |
| 75% DBS+25% N (7.95 tha$^{-1}$+34.5 N kgha$^{-1}$) | 5.3 | 0.14 | 5.4 | 24.4 | 11.63 | 1.7 |
| 50% DBS+50% N (5.3 tha$^{-1}$+69 N kgha$^{-1}$) | 5.3 | 0.14 | 4.5 | 26.7 | 11.61 | 1.6 |
| 25% DBS+75% N (2.65 tha$^{-1}$+103.5 N kgha$^{-1}$) | 5.2 | 0.14 | 4.6 | 27.5 | 11.59 | 1.7 |
| 100% DBS+25% N (10.6 tha$^{-1}$+34.5 N kgha$^{-1}$) | 5.4 | 0.17 | 4.4 | 28.4 | 11.59 | 1.9 |
| 100% DBS+50%N (10.6 tha$^{-1}$+69 N kgha$^{-1}$) | 5.4 | 0.16 | 4.4 | 28.5 | 11.61 | 1.9 |
| **LSD** | **NS** | **NS** | **NS** | **NS** | **NS** | **NS** |
| **CV%** | **2.12** | **13.5** | **21.6** | **9.0** | **0.17** | **13.5** |
| 2020–21 (Y2S2) | | | | | | |
| Treatment | pH | TN% | Av P (ppm) | CEC (cmol kg$^{-1}$) | C: N | OC% |
| Control (0,0) | 5.3[cd] | 0.17 | 12.2 | 30.6 | 12.10 | 2.1 |
| RN (138 N kgha$^{-1}$) | 5.4b[cd] | 0.17 | 7.1 | 31.9 | 11.60 | 1.9 |
| 50% DBS (5.3 tha$^{-1}$) | 5.5[abc] | 0.18 | 11.5 | 30.4 | 11.60 | 2.0 |
| 100% DBS (10.6 tha$^{-1}$) | 5.6[ab] | 0.18 | 9.5 | 32.1 | 12.62 | 2.3 |
| 75% DBS (7.95 tha$^{-1}$) | 5.6[ab] | 0.17 | 13.2 | 32.1 | 12.00 | 2.0 |
| 75% DBS+25% N (7.95 tha$^{-1}$+34.5 N kgha$^{-1}$) | 5.5[abc] | 0.18 | 13.3 | 33.5 | 11.60 | 2.1 |
| 50% DBS+50% N (5.3 tha$^{-1}$+69 N kgha$^{-1}$) | 5.5[abcd] | 0.17 | 13.1 | 32.6 | 12.01 | 2.1 |
| 25% DBS+75% N (2.65 tha$^{-1}$+103.5 N kgha$^{-1}$) | 5.3[d] | 0.16 | 9.5 | 30.9 | 11.60 | 1.9 |
| 100% DBS+25% N (10.6 tha$^{-1}$+34.5 N kgha$^{-1}$) | 5.6[a] | 0.18 | 11.6 | 31.8 | 11.80 | 2.1 |
| 100% DBS+50% N (10.6 tha$^{-1}$+69 N kgha$^{-1}$) | 5.4[bcd] | 0.17 | 15.3 | 33.4 | 11.60 | 2.1 |
| **LSD** | **2.3** | **NS** | **NS** | **NS** | **NS** | **NS** |
| **CV%** | **0.22** | **9.9** | **29.1** | **6.1** | **5.9** | **13.8** |

Means followed by the same letter (s) within the column are not significantly different at ($P \leq 0.05$). control = is no nitrogen and dry bio slurry, RNP = percent of recommended nitrogen and phosphorus, DBS = dry bio-slurry, pH = power of hydrogen concentration, TN% = total nitrogen percent, AvP = available phosphorus, OC% = organic carbon percent, C: N ratio = carbon to nitrogen ratio, CEC = cation exchange capacity, OM% = organic matter percent, Y1S1 = year one site one, Y2S1 = year two site one and Y2S2 = year two site two.

**Table 4. The residual effect of dry bio-slurry on soil chemical properties after wheat harvesting in the potato-wheat cropping system.**

| Treatment | pH | TN% | Av P (ppm) | CEC (cmol kg$^{-1}$) | C: N | OC% |
|---|---|---|---|---|---|---|
| **2020–21 (Y1S1)** | | | | | | |
| Control (0,0) | 5.40 | 0.174bcd | 8.97 | 21.39 | 9.68 | 1.69 |
| RN (138 N kgha$^{-1}$) | 5.40 | 0.196a | 10.12 | 21.69 | 10.01 | 1.95 |
| 50% DBS (5.3 tha$^{-1}$) | 5.40 | 0.197a | 9.25 | 26.13 | 9.64 | 1.88 |
| 100% DBS (10.6 tha$^{-1}$) | 5.38 | 0.168d | 11.04 | 23.51 | 11.81 | 1.98 |
| 75% DBS (7.95 tha$^{-1}$) | 5.40 | 0.192ab | 9.34 | 25.60 | 9.42 | 1.82 |
| 75% DBS+25% N (7.95 tha$^{-1}$+34.5 N kgha$^{-1}$) | 5.42 | 0.172cd | 9.07 | 24.51 | 11.98 | 2.06 |
| 50% DBS+50% N (5.3 tha$^{-1}$+69 N kgha$^{-1}$) | 5.36 | 0.188abc | 10.06 | 22.09 | 9.61 | 1.80 |
| 25% DBS+75% N (2.65 tha$^{-1}$+103.5 N kgha$^{-1}$) | 5.35 | 0.202a | 10.08 | 25.11 | 10.14 | 2.04 |
| 100% DBS+25% N (10.6 tha$^{-1}$+34.5 N kgha$^{-1}$) | 5.31 | 0.194ab | 10.46 | 21.81 | 11.01 | 2.13 |
| 100% DBS+50% N (10.6 tha$^{-1}$+69 N kgha$^{-1}$) | 5.37 | 0.205a | 10/76 | 23.78 | 9.35 | 1.89 |
| **LSD** | NS | 0.020 | NS | NS | NS | NS |
| **CV%** | 1.3 | 6.2 | 12.7 | 11.9 | 14.8 | 13.3 |
| **2021–22 (Y2S1)** | | | | | | |
| Treatment | pH | TN% | Av P (ppm) | CEC (cmol kg$^{-1}$) | C: N | OC% |
| Control (0,0) | 5.61 | 0.178 | 10.15 | 27.70d | 7.61d | 1.36b |
| RN (138 N kgha$^{-1}$) | 5.53 | 0.179 | 10.22 | 30.52bc | 10.31abc | 1.85a |
| 50% DBS (5.3 tha$^{-1}$) | 5.69 | 0.180 | 11.56 | 34.67a | 12.25abc | 2.20a |
| 100% DBS (10.6 tha$^{-1}$) | 5.67 | 0.160 | 9.85 | 31.77bc | 12.42ab | 1.98a |
| 75% DBS (7.95 tha$^{-1}$) | 5.64 | 0.194 | 9.85 | 31.09bc | 9.74cd | 1.88a |
| 75% DBS+25% N (7.95 tha$^{-1}$+34.5 N kgha$^{-1}$) | 5.70 | 0.178 | 11.76 | 29.56cd | 10.79abc | 1.90a |
| 50% DBS+50% N (5.3 tha$^{-1}$+69 N kgha$^{-1}$) | 5.62 | 0.167 | 10.69 | 30.63bc | 11.38abc | 1.90a |
| 25% DBS+75% N (2.65 tha$^{-1}$+103.5 N kgha$^{-1}$) | 5.54 | 0.186 | 9.59 | 30.37bc | 10.00bcd | 1.86a |
| 100% DBS+25% N (10.6 tha$^{-1}$+34.5 N kgha$^{-1}$) | 5.58 | 0.172 | 9.88 | 31.99bc | 12.82a | 2.20a |
| 100% DBS+50% N (10.6 tha$^{-1}$+69 N kgha$^{-1}$) | 5.44 | 0.185 | 10.19 | 32.91ab | 11.78abc | 2.17a |
| **LSD** | NS | NS | NS | 2.60 | 2.61 | 0.41 |
| **CV%** | 2.0 | 6.7 | 14.4 | 4.8 | 14.1 | 12.6 |
| **2021–22 (Y2S2)** | | | | | | |
| Treatment | pH | TN% | Av P (ppm) | CEC (cmol kg$^{-1}$) | C: N | OC% |
| Control (0,0) | 5.51 | 0.176de | 10.03 | 29.43c | 12.03 | 2.12 |
| RN (138 N kgha$^{-1}$) | 5.65 | 0.190bcd | 8.76 | 30.69bc | 12.19 | 2.29 |
| 50% DBS (5.3 tha$^{-1}$) | 5.62 | 0.186bcde | 13.49 | 35.92 a | 12.78 | 2.38 |
| 100% DBS (10.6 tha$^{-1}$) | 5.62 | 0.209ab | 14.08 | 33.68ab | 11.63 | 2.43 |
| 75% DBS (7.95 tha$^{-1}$) | 5.76 | 0.178cde | 11.74 | 34.47a | 13.41 | 2.39 |
| 75% DBS+25% N (7.95 tha$^{-1}$+34.5 N kgha$^{-1}$) | 5.68 | 0.207ab | 12.93 | 34.81a | 12.29 | 2.54 |
| 50% DBS+50%N (5.3 tha$^{-1}$+69 N kgha$^{-1}$) | 5.57 | 0.202abc | 13.08 | 35.88a | 11.47 | 2.30 |
| 25% DBS+75% N (2.65 tha$^{-1}$+103.5 N kgha$^{-1}$) | 5.59 | 0.164e | 11.07 | 33.27ab | 13.52 | 2.19 |
| 100% DBS+25% N (10.6 tha$^{-1}$+34.5 N kgha$^{-1}$) | 5.66 | 0.223a | 11.51 | 36.06a | 10.09 | 2.24 |
| 100% DBS+50% N (10.6 tha$^{-1}$+69 N kgha$^{-1}$) | 5.66 | 0.207ab | 11.21 | 35.95a | 11.75 | 2.42 |
| **LSD** | NS | 0.025 | NS | 3.78 | NS | NS |
| **CV%** | 2.3 | 7.6 | 17.8 | 6.5 | 10.5 | 7.3 |

Means followed by the same letter (s) within the column are not significantly different at ($P \leq 0.05$). RNP = percent of recommended nitrogen and phosphorus, DBS = dry bio-slurry, pH = power of hydrogen concentration, TN% = total nitrogen percent, AvP = available phosphorus, OC% = organic carbon percent, C: N ratio = carbon to nitrogen ratio, CEC = cation exchange capacity, OM% = organic matter percent, Y1S1 = year one site one, Y2S1 = year two site one and Y2S2 = year two site two.

an application of 70cm$^3$ scored the highest (1.36%) total nitrogen as compared to control which was given 0.07%.

On the other hand, OC, C: N and CEC were significantly affected at Y2S1 (Table 4), While; in Y2S2 only CEC is significant at (p < 0.05) Table 4. Based on this; numerically the highest value of CEC (34.6), C: N (12.82) & OC% (2.20) was observed by application of 50% dry bio-slurry (DBS) and 100% dry bio-slurry (DBS) with 25% RN as compared to control on Y2S1.On Y2S2 the maximum value of CEC (cmolkg$^{-1}$) 36.06 was obtained in plots that receive 100% DBS with 25%N than control. This might be the positive effect of applied DBS on the improvement of organic matter and soil holding capacity of positive cations on its exchangeable site. The finding agreed with [33] who revealed that an application of 41.3 m$^{-3}$ liquid bio-slurry with 20.5 kg ha$^{-1}$ N significantly increased soil organic carbon than untreated plots. The study conducted by [34] also indicated that an application of 10 tha$^{-1}$ compost with 50% N kg ha$^{-1}$ significantly increased OC% by scoring the maximum value of 0.67 than the control treatment. Organic amendments significantly enhanced SOC and they have had a considerable effect on soil microbes' abundance and nutrient availability and uptake this may alter the C: N ratio of the soil. However, the addition of external organic matter with a low C: N ratio may induce the mineralization of OM. This makes nitrogen trapped by organic matter; a phenomenon known as the priming effect [35]. The application of the 50% DBS and 100% DBS with 25% N gives the highest CEC (34.67 & 36.06) at Y2S1 and Y2S2 respectively as compared to the control (Table 4). Such increment in CEC might be due to the application of DBS on soil, which makes the colloidal site of the soil negatively charged for storing of basic cations. The finding agreed with [36] who reported that the use of organic farmyard manure (FYM) and inorganic fertilizers significantly increased CEC over the control.

## 3.3 Effects of dry bio slurry and nitrogen on yield and yield components of potato and wheat

### 3.3.1 Plant height and number of Stem per hill of potato.
Integration of dry bio slurry with nitrogen (N) significantly (P < 0.05) affects plant height (Table 5). Numerically, the highest values of plant height (52.1 and 48.8 cm) were achieved with the addition of recommended

**Table 5. Main and residual effects of dry bio-slurry with equivalence N on growth parameters of potato and wheat.**

| Treatments | Main effect of DBS on Potato | | Residual effect DBS on Wheat | |
|---|---|---|---|---|
| | PH (cm) | NSPH | PH (cm) | SL (cm) |
| Control (0,0) | 29.4$^{de}$ | 3.3 | 68.2d | 6.7d |
| RN (138 N kgha$^{-1}$) | 52.1$^a$ | 4.8 | 89.1c | 8.5c |
| 50% DBS (5.3 tha$^{-1}$) | 28.2$^e$ | 4.4 | 92.3ab | 8.7bc |
| 100% DBS (10.6 tha$^{-1}$) | 30.4$^{de}$ | 4.4 | 93.2a | 9.3a |
| 75% DBS (7.95 tha$^{-1}$) | 36.6$^{cd}$ | 4.5 | 93.0ab | 9.0abc |
| 75% DBS+25% N (7.95 tha$^{-1}$+34.5 N kgha$^{-1}$) | 41.9$^{bc}$ | 4.6 | 92.1abc | 9.0abc |
| 50% DBS+50% N (5.3 tha$^{-1}$+69 N kgha$^{-1}$) | 42.7$^{bc}$ | 3.7 | 90.2bc | 8.9abc |
| 25% DBS+75% N (2.65 tha$^{-1}$+103.5 N kgha$^{-1}$) | 48.8$^{ab}$ | 4.0 | 90.2bc | 8.7bc |
| 100% DBS+25% N (10.6 tha$^{-1}$+34.5 N kgha$^{-1}$) | 44.0$^{bc}$ | 4.3 | 92.1ab | 9.0abc |
| 100% DBS+50% N (10.6 tha$^{-1}$+69 N kgha$^{-1}$) | 47.1$^{ab}$ | 4.2 | 90.6abc | 9.1ab |
| **LSD** | **7.9** | **NS** | 3.0 | 0.5 |
| **CV%** | **20.9** | **30.1** | 3.6 | 6.6 |

Means followed by the same letter (s) within the column are not significantly different at (P ≤ 0.05). DBS = dry bio slurry, N = nitrogen, PH = plant height, NSPH = number of stems per hill, SL = spike length.

nitrogen (RN) and 25% DBS with 75% N respectively as compared to the control (Table 5). Increasing plant height in response to DBS with N fertilizer may be due to the improvement of physico—chemical properties of the soil in terms of water absorption for nutrient utilization of the plant. Moreover, DBS may deliver balanced micro and macronutrients as well as it enhance the availability of plant nutrients, which would help to speed up the metabolic activity of micro-organisms and promote plant growth. The result agreed with the findings of [37] who observed the longer plants when potatoes were applied with farmyard manure (13.5 tha$^{-1}$) and NPS (245.1 kgha$^{-1}$).lt is also harmonized with the findings of [38] who recorded the maximum plant height of french bean from the application of 120 kg N ha$^{-1}$ while, the minimum value was obtained from the control treatment. Another study that was conducted by [39] also, reflected that the highest value of mung bean plant height (78.08 cm) was recorded from the treatment which received 20:50 NP kg ha$^{-1}$ with inoculation of Rhizobium as compared to the lowest value of 68 cm on the control treatment. On the other hand, a combined analysis of variance revealed that DBS with N fertilizers had no significant effects on the stem numbers of potato per hill (Table 5). This might be the parameter favored for genetic makeup, physiological age and tuber seed size rather than a nutrient supplement. This finding lined with the study of [40] who observed that the shoot number of potatoes is mostly determined by the genetic makeup, the physiological age, and the size of potato seed tubers rather than mineral nutrients added in the form of fertilizer.

**3.3.2 Plant height and spike length of wheat.**   Over the years residual analysis of dry-bio slurry on wheat indicated that it significantly affected both plant height and spike length at (P < 0.05) in (Table 5). The highest values of plant height and spike length (93.2 and 9.3 cm) were achieved by the application of 100% DBS as compared to untreated plots (Table 5). This might be from the positive effect of DBS for delivering balanced micro and macronutrients by enhancing the availability of plant nutrients via improved soil properties. The result agreed with the findings of [41] who observed that the longer plants in plots that received 75% cow dung with 25% vermi-compost than untreated plots. lt is also harmonized with the findings of [42] who recorded that the maximum plant height (101.5 cm) value of Boro rice from the application of 5 tha$^{-1}$ tricho-compost as compared to the control that was scored the minimum value (78.6 cm). Similarly, the result of spike length also much to the study of [43] who said that an application of 15 tha$^{-1}$ biogas slurry gives the highest spike length of wheat than the checked treatment or control.

**3.3.3 Number of tubers per plant and total tuber yield of potato.**   The combined analysis of results across years and sites indicated that the yield and yield component of potato signifi-cantly differed at (P < 0.05) by the effects of DBS and RN (Table 6). The application of 75% N with 25% DBS gives the highest fresh total tuber yield (27.6 tha$^{-1}$) while the lowest fresh tuber yield (8.6 tha$^{-1}$) was observed at control. This might be due to the releasement of nitrogen (N) from dry bio-slurry (DBS) and Urea to soil solution that makes for plant better growth and development. Moreover, it could be due to the addition of both macro and micronutrients from the dry bio-slurry (DBS) by improving soil pH, organic carbon, total nitrogen phospho-rus and cation exchange capacity. This study is in line with the findings of [13] who revealed that the supplying of recommended inorganic fertilizer (100kg DAP, 50kg Urea and 50kg Murate potash per hectare) with 8 tha$^{-1}$ bio-slurry gives a maximum (266.7 tha$^{-1}$) yield of cab-bage as compared to the lowest (160 tha$^{-1}$) from the control treatment. It gives about 66.7% yield increment due to the combination of both bio-slurry and recommended fertilizers over control. On the other hand, the study done by [44] indicated that the lowest value of fresh shoot biomass and marketable yield of potato tuber was achieved from control while the high-est values were obtained in plots that treated combined application of farmyard manure and recommended nitrogen and phosphorus.

Similarly, a number of tubers per plant (NTPP) was significantly affected at (P < 0.05) by the application of DBS with equivalence nitrogen. The Maximum value of NTPP was observed

**Table 6. Response of potato and wheat yield parameters for dry bio slurry and equivalence nitrogen as a main and residual effect.**

| Treatments | Main effect of DBS on Potato | | Residual effect DBS on Wheat | | |
|---|---|---|---|---|---|
| | NTPP | TYD t ha$^{-1}$ | GY (t ha$^{-1}$) | BY (tha$^{-1}$) | 1000 SW(g) |
| Control (0,0) | 4.2[e] | 8.6[d] | 1.23d | 3.11d | 30.1 |
| RN (138 N kgha$^{-1}$) | 10.9[a] | 26.2[ab] | 3.17c | 7.9c | 32.4 |
| 50% DBS (5.3 tha$^{-1}$) | 5.0[de] | 12.0[cd] | 3.33bc | 8.37bc | 32.8 |
| 100% DBS (10.6 tha$^{-1}$) | 6.4[cd] | 13.4[c] | 3.85a | 9.52ab | 33.6 |
| 75% DBS (7.95 tha$^{-1}$) | 7.3[bc] | 14.9[c] | 3.83a | 9.59a | 33.9 |
| 75% DBS+25% N (7.95 tha$^{-1}$+34.5 N kgha$^{-1}$) | 6.5[bcd] | 23.0[b] | 3.47abc | 9.08ab | 33.2 |
| 50%DBS+50% N (5.3 tha$^{-1}$+69 N kgha$^{-1}$) | 6.4[cd] | 24.6[ab] | 3.47abc | 8.53abc | 32.1 |
| 25% DBS+75% N (2.65 tha$^{-1}$+103.5 N kgha$^{-1}$) | 6.8[bc] | 27.6[a] | 3.28bc | 8.43abc | 32.6 |
| 100% DBS+25% N (10.6 tha$^{-1}$+34.5 N kgha$^{-1}$) | 7.0[bc] | 25.2[ab] | 3.70ab | 9.56a | 33.0 |
| 100% DBS+50% N (10.6 tha$^{-1}$+69 N kgha$^{-1}$) | 8.1[b] | 26.8[a] | 3.59abc | 9.06abc | 32.4 |
| **LSD** | 1.7 | 3.7 | 0.5 | 1.2 | NS |
| **CV%** | 25.6 | 19.4 | 15.6 | 14.8 | 10.1 |

Means followed by the same letter (s) within the column are not significantly different at (P ≤ 0.05). DBS = dry bio slurry, N = nitrogen, PH = plant height, NTPP = number of tubers per plant, TYD, = total tuber yield, GY = grain yield, BY = above ground biomass and SW = seed weight.

through the application of the recommended NP as compared to control treatment (Table 6). Even if the maximum value occurs at the recommended NP, the combination of both dry bio slurry and nitrogen treatment also gives a better yield advantage than the control. This might be due to; the harmonization of organic and inorganic fertilizers for the uptake and assimilation of nutrients to potato tubers by increasing the availability of native soil nutrients through higher biological activities. The result coincides with the study of [45] who observed that an application of organic and inorganic fertilizers increased the number of tubers per plant in treated plots than control or untreated plots. Another study done by [31] showed that an application of dry bio-slurry with recommended nitrogen and phosphorus can increase the number of fruits per plant of tomato up to 40 to 73% than the control.

**3.3.4 Grain yield and aboveground biomass of wheat.** Yield and above ground biomass of wheat were significantly affected at (P < 0.05) in (Table 6) by the residual effect of dry bio-slurry. The application of 100% DBS gives the highest grain yield (3.85 tha$^{-1}$) as compared to control. Similarly, the addition of 75% DBS gives the highest above ground biomass of wheat (9.59 tha$^{-1}$) compared to the control (Table 6). Releasing N from dry bio-slurry (DBS) to soil solution may contribute to the plants better growth and development. Moreover, it could be due to the addition of both macro and micronutrients from the dry bio-slurry into the soil solution (rhizosphere) by increasing the availability of native soil nutrients. The result coincides with the findings of [43] who reported that the supplement of 10 and 15 tha$^{-1}$ of bio-slurry significantly increased the grain yield of wheat as compared to the control. Similarly, a study conducted by [46] revealed that the application of both biogas slurry and chemical fertilizer at 50% has a good strategy for sustainable crop yield production and soil health improvement. Similarly, the study conducted by [47] also showed that an addition of 100% dry bio-slurry significantly increased the stover and stalk yield of maize by 45.5 and 42.2% respectively to control treatment via improving soil biological activities.

## 3.4 Economic analysis

Net benefits were done by calculating a current fertilizer (Urea) cost of 0.27 US$ kg$^{-1}$, the cost of DBS kg$^{-1}$ was 0.004 US$, the field price of a potato was 0.12 US$ kg$^{-1}$ and the cost of labor

**Table 7.  Partial budget and marginal analysis of potato as affected by the application of dry bio-slurry with nitrogen at Yilemana Densa District.**

| Treatments (RN +DBS kgha⁻¹) | 10% Adjusted tuber Yield t ha⁻¹ | Total variable Cost US$ha⁻¹ | Net Benefits US$ ha⁻¹ | MRR% |
|---|---|---|---|---|
| Control | 7.74 | 0 | 928.8 | - |
| 50% DBS | 10.8 | 35.2 | 1260.8 | 943.2 |
| 75% DBS | 13.41 | 52.8 | 1556.4 | 1679.6 |
| 100% DBS | 12.06 | 70.4 | 1376.8 | D |
| 50% DBS+50% N | 22.14 | 84.5 | 2572.3 | 3204.7 |
| 25% DBS+75% N | 24.84 | 91.6 | 2889.2 | 4463.3 |
| 100% DBS+25% N | 22.68 | 95.1 | 2626.5 | D |
| RNP | 23.58 | 98.7 | 2730.9 | D |
| 100% DBS + 50% N | 24.12 | 119.7 | 2774.7 | D |
| 75% DBS + 25% N | 20.7 | 217.5 | 2266.5 | D |

RNP = percent of recommended nitrogen and phosphorus in kg per hectare, DBS = dry bio-slurry in kilogram per hectare, MRR = marginal rate of return; D = is dominated treatments

per man day in the area was 1.4 US$. The marginal rate of return of 100% was used to determine the acceptability of treatments. The economic analysis indicated that most treatments give a higher net benefit than the control (Table 7). The addition of 25% DBS with 75% RN gave 2889.2 US$ net benefit with a 4463.3% marginal rate of return. This replied that for every 1 US$ invested for 25% DBS with 75% RN in the field, farmers can obtain additional 44.633 US$ [25]. Undominated treatment rates could be acceptable for potato producers in the study area except for the dominated treatments Therefore, the most economical rate for producers with low cost and higher benefits was 25% DBS with 75% RN.

## 4. Conclusion and recommendation

The main and residual effects of dry bio-slurry had a considerable impact on a potato and wheat yield and yield components. In comparison to the control, both crops' productivity increased with the substitution of dry bio-slurry and nitrogen. So, use of 25% dry bio-slurry (DBS) with 75% recommended nitrogen (RN) could be promoted for the production of potatoes and wheat in the study area. For the future, similar studies should be done across locations and crops, in permanent plots for sustainable crop production and soil health enhancement.

## Acknowledgments

The author is grateful to Adet Agricultural research center for its administrative and budget support.

## Author Contributions

**Conceptualization:** Zelalem Addis, Tadele Amare, Anteneh Abewa, Tesfaye Feyisa.

**Data curation:** Zelalem Addis, Tesfaye Feyisa, Abere Tenagne.

**Formal analysis:** Zelalem Addis, Abrham Awoke.

**Methodology:** Zelalem Addis, Bitewlgn Kerebih, Abrham Awoke.

**Project administration:** Abere Tenagne.

**Resources:** Bitewlgn Kerebih.

**Software:** Zelalem Addis.

**Supervision:** Tadele Amare, Anteneh Abewa.

**Validation:** Zelalem Addis.

**Visualization:** Zelalem Addis.

**Writing – original draft:** Zelalem Addis.

**Writing – review & editing:** Tadele Amare.

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
