## [Decision Letter · Decision Letter 0]

9 May 2023

PONE-D-23-10312Main and residual effect of dry bio-slurry with nitrogen fertilizer application on soil properties and crop production under Potato-wheat cropping systemPLOS ONE

Dear Dr. Musse,

Thank you for submitting your manuscript to PLOS ONE. After careful consideration, we feel that it has merit but does not fully meet PLOS ONE’s publication criteria as it currently stands. Therefore, we invite you to submit a revised version of the manuscript that addresses the points raised during the review process.

We look forward to receiving your revised manuscript.

Kind regards,

Ravinder Kumar, Ph.D.

Academic Editor

PLOS ONE

Journal Requirements:

"NO - Include this sentence at the end of your statement: The funders had no role in study design, data collection and analysis, decision to publish, or preparation of the manuscript."

"NO - Include this sentence at the end of your statement: The funders had no role in study design, data collection and analysis, decision to publish, or preparation of the manuscript."

Reviewers' comments:

Reviewer's Responses to Questions

**Comments to the Author**

1. Is the manuscript technically sound, and do the data support the conclusions?

Reviewer #1: Yes

Reviewer #2: Yes

Reviewer #3: Partly

Reviewer #4: Partly

2. Has the statistical analysis been performed appropriately and rigorously? 

Reviewer #1: Yes

Reviewer #2: Yes

Reviewer #3: Yes

Reviewer #4: Yes

3. Have the authors made all data underlying the findings in their manuscript fully available?

Reviewer #1: Yes

Reviewer #2: Yes

Reviewer #3: No

Reviewer #4: No

4. Is the manuscript presented in an intelligible fashion and written in standard English?

Reviewer #1: No

Reviewer #2: No

Reviewer #3: No

Reviewer #4: No

5. Review Comments to the Author

Reviewer #1: Review report

Manuscript number: PONE-D-23-10312

Title: Main and residual effect of dry bio-slurry with nitrogen fertilizer application on soil properties and crop production under Potato-wheat cropping system

General comment

Integrated application of organic and inorganic fertilizers is an important soil amendment strategy that improves the sustainability of crop production including potato and wheat. Potato and wheat are produced by smallholder farmers in Ethiopia where enhancing their productivity contributes a lot for the livelihood improvement and ensuring food security in the country. In this regard, this research work may have significant contribution.

However, the manuscript is not well structured and lacks justification and objectives. The methodology is not well described and the results are not well presented and discussed. The authors are advised to make use of Microsoft grammar and spelling check. Comments to the individual section of the manuscript are presented below.

Title: It is suggested to rephrase as “Effects of dry-bio-slurry supplemented with chemical nitrogen fertilizer on growth and yield of potato and wheat under Potato-wheat cropping system

Abstract

• It is recommended to include problem statement and objective of the study at the beginning of the abstract.

• The conclusion should be changed after correcting the MRR analysis in the results and discussion.

Introduction

• The authors should justify why intergraded application of organic and inorganic fertilizer is better than the solo application of the fertilizers.

• Previous research experiences in the integrated application of fertilizers should be reviewed.

• Most of the information/references are very old and the authors should change them with recent references

• There are sentences that are difficult to understand and unfinished, which should be revised

• A lot of grammatical and typographical errors that should be revised

• For more information please refer the commented document

Methodology

This part of the manuscript should be given due attention and requires rigorous revision. Description of the study area, the production system and data collection methods implemented (growth and yield) should be clearly described so that it is understandable and could be repeated by other person (for more information, please refer the attached document).

Results and discussion

The way results are presented is not attractive for the reader. It is suggested to present first the results and then forward why it could be and then compare the results with others.

Results should be presented in the past tense form.

Although no description of site 1 and site 2 in the methodology, results of site 1 and site 2 (soil sample results) are presented, which are confusing.

The economic analysis should be done once again as the MRR of 50%DBS+50%N (8461.1%) is not correct. Accordingly, the conclusion and recommendation part should be revised based on the corrected MRR value.

For more information, refer commented document

Conclusion and recommendation

It should be revised after the revision of the MRR values.

References

The way reference listed should be uniform and meet the guideline of the journal.

Reviewer #2: I must appreciate authors for nice work in which they have tried to understand supplementation of N with DBS on the performance of Potato-wheat cropping system and soil quality. The study has been conducted by following established procedures and methods and is scientifically sound. The data recorded and presented in the MS are also in order. I have thoroughly gone through the MS and realized that there is still scope to further improve its quality, presentation and content. According some suggestions/comments have been tendered on the MS itself. These comments are easy to understand and self explanatory. In addition to these comments, some specific comments are as under. Authors are advised to revise the MS by duly incorporating all the suggestions/comments.

1. English language polishing is highly needed to bring the information of the MS in presentable form.

2. Authors are advised to include more yield attributes of both the crops for better and clear understanding of treatment effects on test crops.

In addition to the above, it is recommended to include system equivalent yield as study has been attempted in system mode.

3. Include recent and updated works to support findings. Authors are also advised to cross check the work cited in text and presented in the reference section and vice versa.

Reviewer #3: The research holds importance in the area where its being conducted. Although the program is well planned but it needs to be re written. The introduction is very weak, does not fully justify the topic. It needs to be revised keeping in view all the important parameters in the title. The results too are poorly written and discussed with most of the incomplete sentences.

the abstract also needs to be rewritten.

Reviewer #4: Starting from the topic need major and careful modifications. The abstract does not contain the mail elements and it is poor. The result section is not properly interpreted. conclusion and recommendation section are poor. Majority of the references are very old.

6. PLOS authors have the option to publish the peer review history of their article (what does this mean?). If published, this will include your full peer review and any attached files.

Reviewer #1: No

Reviewer #2: **Yes: **Sushil Kumar

Reviewer #3: No

Reviewer #4: No

---

## [Author Response · Author response to Decision Letter 0]

23 Jun 2023

Reviewer 1

.title: It is suggested to rephrase as “Effects of dry-bio-slurry supplemented with chemical nitrogen fertilizer on growth and yield of potato and wheat under Potato-wheat cropping system

We have are adjust the title based on reviewer “Effects of dry-bio-slurry supplemented with chemical nitrogen fertilizer on growth and yield of potato and wheat under Potato-wheat cropping system 

Abstract

• It is recommended to include problem statement and objective of the study at the beginning of the abstract.

• The conclusion should be changed after correcting the MRR analysis in the results and discussion.

Here we also incorporate statement of the problem at the begging of abstract 

we have tried to check MRR by back tracing of the data its correct but our recommendation and conclusion adjusted from 50%RN+ 50DBS to 25%DBS +75%RN 

Introduction

• The authors should justify why intergraded application of organic and inorganic fertilizer is better than the solo application of the fertilizers.

Accepted and incorporated in main document (manuscript)

• Previous research experiences in the integrated application of fertilizers should be reviewed.

Accepted and incorporated in main document (manuscript)

• Most of the information/references are very old and the authors should change them with recent references

in case of reference we have tried to explorer recent works and adjust it based on the comment 

• There are sentences that are difficult to understand and unfinished, which should be revised

Accepted and revised in main document (manuscript)

• A lot of grammatical and typographical errors that should be revised we also tried adjust the language errors

For more information please refer the commented document

Methodology

This part of the manuscript should be given due attention and requires rigorous revision. Description of the study area, the production system and data collection methods implemented (growth and yield) should be clearly described so that it is understandable and could be repeated by other person (for more information, please refer the attached document).

Accepted and incorporated as per the comment in main document (manuscript)

Results and discussion

The way results are presented is not attractive for the reader. It is suggested to present first the results and then forward why it could be and then compare the results with others.

Results should be presented in the past tense form.

Although no description of site 1 and site 2 in the methodology, results of site 1 and site 2 (soil sample results) are presented, which are confusing.

The economic analysis should be done once again as the MRR of 50%DBS+50%N (8461.1%) is not correct. Accordingly, the conclusion and recommendation part should be revised based on the corrected MRR value.

For more information, refer commented document

Conclusion and recommendation

It should be revised after the revision of the MRR values.

All the comments are pertinent, and incorporated in main document (manuscript)

References

The way reference listed should be uniform and meet the guideline of the journal. 

Based on journal guide line we adjust it

Reviewer #2:

I must appreciate authors for nice work in which they have tried to understand supplementation of N with DBS on the performance of Potato-wheat cropping system and soil quality. The study has been conducted by following established procedures and methods and is scientifically sound. The data recorded and presented in the MS are also in order. I havethoroughly gone through the MS and realized that there is still scope to further improve its quality, presentation and content. According some suggestions/comments have been tendered on the MS itself. These comments are easy to understand and self explanatory. In addition to these comments, some specific comments are as under. Authors are advised to revise the MS by duly incorporating all the suggestions/comments.

1. English language polishing is highly needed to bring the information of the MS in presentable form.

Accepted and corrected in the main document

2. Authors are advised to include more yield attributes of both the crops for better and clear understanding of treatment effects on test crops.

In addition to the above, it is recommended to include system equivalent yield as study has been attempted in system mode.

Not applicable for rotation system

3. Include recent and updated works to support findings. Authors are also advised to cross check the work cited in text and presented in the reference section and vice versa.

Accepted and incorporated

Reviewer #3: 

The research holds importance in the area where its being conducted. Although the program is well planned but it needs to be re written. The introduction is very weak, does not fully justify the topic. It needs to be revised keeping in view all the important parameters in the title. The results too are poorly written and discussed with most of the incomplete sentences.

the abstract also needs to be rewritten..

All are accepted and incorporated in the main document

---

## [Decision Letter · Decision Letter 1]

10 Jul 2023

PONE-D-23-10312R1Effects of dry-bio-slurry supplemented with chemical nitrogen fertilizer on growth and yield of potato and wheat under Potato-wheat cropping systemPLOS ONE

Dear Dr. Musse,

Thank you for submitting your manuscript to PLOS ONE. After careful consideration, we feel that it has merit but does not fully meet PLOS ONE’s publication criteria as it currently stands. Therefore, we invite you to submit a revised version of the manuscript that addresses the points raised during the review process.

We look forward to receiving your revised manuscript.

Kind regards,

Ravinder Kumar, Ph.D.

Academic Editor

PLOS ONE

Reviewers' comments:

Reviewer's Responses to Questions

**Comments to the Author**

1. If the authors have adequately addressed your comments raised in a previous round of review and you feel that this manuscript is now acceptable for publication, you may indicate that here to bypass the “Comments to the Author” section, enter your conflict of interest statement in the “Confidential to Editor” section, and submit your "Accept" recommendation.

Reviewer #1: (No Response)

Reviewer #2: (No Response)

Reviewer #3: All comments have been addressed

Reviewer #4: (No Response)

2. Is the manuscript technically sound, and do the data support the conclusions?

Reviewer #1: Yes

Reviewer #2: Partly

Reviewer #3: Yes

Reviewer #4: Yes

3. Has the statistical analysis been performed appropriately and rigorously? 

Reviewer #1: Yes

Reviewer #2: Yes

Reviewer #3: Yes

Reviewer #4: Yes

4. Have the authors made all data underlying the findings in their manuscript fully available?

Reviewer #1: Yes

Reviewer #2: (No Response)

Reviewer #3: Yes

Reviewer #4: Yes

5. Is the manuscript presented in an intelligible fashion and written in standard English?

Reviewer #1: No

Reviewer #2: No

Reviewer #3: No

Reviewer #4: No

6. Review Comments to the Author

Reviewer #1: Although most of the comments are addressed, there are still issues to be corrected or revised. The MRR analysis should be done once again which will probably have influences on the conclusion and recommendations of the study. Moreover there are a lot of grammatical and editorial errors that should be corrected. For more detail please refer the commented document.

Reviewer #2: Although authors have took reasonable time and efforts to improve the content and quality of the manuscript by duly incorporating the reviewers comments/suggestions but still so many comments are either not attended properly or superficially considered. The logical flow and presentation of the data in the results and discussion section are still not in order. The clarity in materials and methods are still missing. Many of the suggestions just for example language polishing are not properly addressed. It is recommended and advised not to accept the manuscript in present form. Authors may advised and asked to do more sincere efforts in improving fluency, content, quality and interpretation of the results in logical and scientific manner.

Reviewer #3: Although all the comments raised by the reviewers have been addressed by the authors but still the manuscript has lots of typographical and grammatical mistakes. The authors need to critically go through the manuscript before its final submission.

Reviewer #4: It is better to modify the topic as follow: Effects of dry-bio-slurry and nitrogen fertilizers on potato and wheat yields under Potato-wheat cropping system, Northwest Ethiopia

Language and coherence of sentences are poor and fragmented. So, the language should be written clearly and precisely by language experts.

I found a lot of errors about spacing, grammar, punctuation and others under the whole section of the paper. So, it needs a revision.

There are tables without any caption, for instance, the caption for table 5 is missing.

The discussion is also poor.

The conclusion and recommendation part are shallow and poor.

7. PLOS authors have the option to publish the peer review history of their article (what does this mean?). If published, this will include your full peer review and any attached files.

Reviewer #1: No

Reviewer #2: No

Reviewer #3: No

Reviewer #4: No

---

## [Author Response · Author response to Decision Letter 1]

13 Jan 2024

I have tried to Adress all comments and suggestion that given by you

---

## [Decision Letter · Decision Letter 2]

7 Feb 2024

PONE-D-23-10312R2Effects of dry-bio-slurry supplemented with chemical nitrogen fertilizer on growth and yield of potato and wheat under Potato-wheat cropping systemPLOS ONE

Dear Dr. Musse,

Thank you for submitting your manuscript to PLOS ONE. After careful consideration, we feel that it has merit but does not fully meet PLOS ONE’s publication criteria as it currently stands. Therefore, we invite you to submit a revised version of the manuscript that addresses the points raised during the review process.

We look forward to receiving your revised manuscript.

Kind regards,

Ravinder Kumar, Ph.D.

Academic Editor

PLOS ONE

Additional Editor Comment:

The authors are again requested to kindly address the concern of all previous reviewers. In revised version also two reviewers have raised the issues, and I am also of the same opinion to give another chance to authors to revise the manuscript so that revised manuscript can be considered for publication.

Reviewers' comments:

Reviewer's Responses to Questions

**Comments to the Author**

1. If the authors have adequately addressed your comments raised in a previous round of review and you feel that this manuscript is now acceptable for publication, you may indicate that here to bypass the “Comments to the Author” section, enter your conflict of interest statement in the “Confidential to Editor” section, and submit your "Accept" recommendation.

Reviewer #1: (No Response)

Reviewer #4: (No Response)

2. Is the manuscript technically sound, and do the data support the conclusions?

Reviewer #1: Partly

Reviewer #4: Partly

3. Has the statistical analysis been performed appropriately and rigorously? 

Reviewer #1: Yes

Reviewer #4: Yes

4. Have the authors made all data underlying the findings in their manuscript fully available?

Reviewer #1: Yes

Reviewer #4: Yes

5. Is the manuscript presented in an intelligible fashion and written in standard English?

Reviewer #1: No

Reviewer #4: No

6. Review Comments to the Author

Reviewer #1: Review report

Manuscript number: PONE-D-23-10312R2

Title: Effects of dry-bio-slurry supplemented with chemical nitrogen fertilizer on growth and yield of potato and wheat under Potato-wheat cropping system

General comment

Although some of them are addressed, the majority of the comments raised in the previous version are not well addressed in the revised document. As some of senior researchers are involved in the study, they have to have a look on the document for its improvement.

Some of major issues that require due attentions of the authors:

• The title shall be revised by considering the replacement/substitution of chemical N with DBS

• Statement of the problem in the abstract requires more articulation,

• Please review previous research findings/practices that showed the improvement of growth and yield of crops and soil properties through application of organic and inorganic fertilizers (only general principles are reviewed).

• Correct the net plot area and MRR, which create deviations in the collected data and conclusion and recommendation

• The recommendation in the abstract and in conclusion is different. Please revise and make a similar recommendation in both sections of the manuscript.

• Revise grammatical and editorial errors throughout the document

• For more information, please refer the commented document

Reviewer #4: Comments

The title should be clear and short.

Suggested title: Effects of dry bioslurry with nitrogen fertilizer on yields of potato and wheat under short term crop rotation system

The abstract lacks short background of the problem.

The result section in the abstract is not well written. Better to summarize the key results only.

There are too long sentences in the paper (see the first four lines of the abstract, description of study area), difficult to realize the concept.

The language is generally poor. It should be further improved by subject experts.

There are editorial errors like spacing (2.2.2Gudenie potato variety, 2.2.3TAY variety of Wheat), spelling (List significant difference (LSD).

SAS software version 9.4 is not available during 2002. You have cited the correct releasing year.

In table 2, describe the clay, sand and silt contents and remove repeated rows containing names of soil properties.

7. PLOS authors have the option to publish the peer review history of their article (what does this mean?). If published, this will include your full peer review and any attached files.

Reviewer #1: No

Reviewer #4: No

---

## [Author Response · Author response to Decision Letter 2]

27 Mar 2024

Response for reviewer 1

I. We have been adjusting the title based on reviewer 4 suggestion “Effects of dry bio slurry with nitrogen fertilizer on yields of potato and wheat under short term crop rotation system by considering the comment of reviewer 1

II. Here we are done the re -articulation of for statement of the problem in the abstract part based on the reviewer comment 

III. Based on the comment we in corporate the previous findings that deals the integration effect of organic and inorganic fertilizers on growth and soil properties

IV. Here we correct both MRR and net plot area-based reviewer comment 

V. We have been making it similar in both parts of the manuscript for recommendation

VI. Here we want to appreciate all of the reviewers including reviewer1 for their effort as manuscript be well edited by language writing. by fully considering and give attention for there frequent comment we have correct and check it by fluent speaker and language skilled persons in addition to our correction effort .so the manuscript well edited and corrected in terms of language 

Response for reviewer 4

I. We have been adjusting the title based on reviewer suggestion “Effects of dry bio slurry with nitrogen fertilizer on yields of potato and wheat under short term crop rotation system 

II. we incorporate the background of the problem in abstract part based on this comment (reviewer 4).

III. we are correct it in both parts by summarize.

IV. we make clear and short in most lines and sentences in the manuscript.

V. Here we want to appreciate all of the reviewers including reviewer4 for their effort as manuscript be well edited by language writing. by fully considering and give attention for their frequent comment we have correct and check it by fluent speaker and language skilled persons in addition to our correction effort .so the manuscript well edited and corrected in terms of language.

VI. We correct it by 2014.

VIII. we also correct it based on the comment

---

## [Decision Letter · Decision Letter 3]

30 Apr 2024

PONE-D-23-10312R3Effects of dry-bio-slurry with nitrogen fertilizer on yield of potato and wheat under short term crop rotationPLOS ONE

Dear Dr. Musse,

Thank you for submitting your manuscript to PLOS ONE. After careful consideration, we feel that it has merit but does not fully meet PLOS ONE’s publication criteria as it currently stands. Therefore, we invite you to submit a revised version of the manuscript that addresses the points raised during the review process.

We look forward to receiving your revised manuscript.

Kind regards,

Ravinder Kumar, Ph.D.

Academic Editor

PLOS ONE

Reviewers' comments:

Reviewer's Responses to Questions

**Comments to the Author**

1. If the authors have adequately addressed your comments raised in a previous round of review and you feel that this manuscript is now acceptable for publication, you may indicate that here to bypass the “Comments to the Author” section, enter your conflict of interest statement in the “Confidential to Editor” section, and submit your "Accept" recommendation.

Reviewer #1: (No Response)

Reviewer #4: (No Response)

2. Is the manuscript technically sound, and do the data support the conclusions?

Reviewer #1: Yes

Reviewer #4: Partly

3. Has the statistical analysis been performed appropriately and rigorously? 

Reviewer #1: Yes

Reviewer #4: Yes

4. Have the authors made all data underlying the findings in their manuscript fully available?

Reviewer #1: Yes

Reviewer #4: Yes

5. Is the manuscript presented in an intelligible fashion and written in standard English?

Reviewer #1: No

Reviewer #4: No

6. Review Comments to the Author

Reviewer #1: (No Response)

Reviewer #4: The authors have made efforts to address the comments given. But there are still comments that require the authors to amend in order to further improve the manuscript.

Comments:

Suggested title: Effects of dry bioslurry and nitrogen fertilizer on potato and wheat yields under

rotation cropping system

The abstract is not clear and precise. Please rewrite this?

The language is also still poor such as grammar, punctuations (line 79: (13) Due), spacing (line 27: 3.85tha-1).

Line 14 and 106: The experiment was contain ten levels of treatments Control ---?

Line 21-22: the results were indicated that; an application of dry bio-slurry with nitrogen fertilizer was ---?

Line 14-21: Write full description of all abbreviations mentioned first in the text: DBS, RN, RCBD, ANOVA, SAS

Most sentences are too long and difficult to understand for readers (Line 9-12, 93-96). So, make the sentences short and clear.

Words found in the title cannot be a keyword like dry bio-slurry, potato, wheat.

Line 96: Location of the study area (110 21' 22'' N and 370 25' 43'' E), is not correct. It is a point, not a polygon?

Amend citations based on the journal guidelines.

Line 93-94: three sites (Y1S1, Y2S1and Y2S2), this not clearly described whether 2 or 3?

Conclusion part is poor. So, rewrite this properly.

7. PLOS authors have the option to publish the peer review history of their article (what does this mean?). If published, this will include your full peer review and any attached files.

Reviewer #1: No

Reviewer #4: No

---

## [Author Response · Author response to Decision Letter 3]

29 May 2024

Dear respected reviewers we really happy and thanks for your hard work and effort to correct this manuscript with your constructive comments. Please respected reviewers as you know this manuscript take long time, so please help us publish on time as much as possible

---

## [Decision Letter · Decision Letter 4]

11 Jun 2024

PONE-D-23-10312R4Effects of dry-bio-slurry with nitrogen fertilizer on yield of potato and wheat under short term crop rotationPLOS ONE

Dear Dr. Musse,

Thank you for submitting your manuscript to PLOS ONE. After careful consideration, we feel that it has merit but does not fully meet PLOS ONE’s publication criteria as it currently stands. Therefore, we invite you to submit a revised version of the manuscript that addresses the points raised during the review process.

We look forward to receiving your revised manuscript.

Kind regards,

Ravinder Kumar, Ph.D.

Academic Editor

PLOS ONE

Journal Requirements:

Reviewers' comments:

Reviewer's Responses to Questions

**Comments to the Author**

1. If the authors have adequately addressed your comments raised in a previous round of review and you feel that this manuscript is now acceptable for publication, you may indicate that here to bypass the “Comments to the Author” section, enter your conflict of interest statement in the “Confidential to Editor” section, and submit your "Accept" recommendation.

Reviewer #1: (No Response)

2. Is the manuscript technically sound, and do the data support the conclusions?

Reviewer #1: (No Response)

3. Has the statistical analysis been performed appropriately and rigorously? 

Reviewer #1: (No Response)

4. Have the authors made all data underlying the findings in their manuscript fully available?

Reviewer #1: (No Response)

5. Is the manuscript presented in an intelligible fashion and written in standard English?

Reviewer #1: No

6. Review Comments to the Author

Reviewer #1: The authors did not still address the MRR calculation correctly. Although the grammar, sentence construction and typographical errors are improved, there are still issues to be addressed in this regard. I still request the authors as well as the editors to take care of grammar and typographical errors.

7. PLOS authors have the option to publish the peer review history of their article (what does this mean?). If published, this will include your full peer review and any attached files.

Reviewer #1: No

---

## [Author Response · Author response to Decision Letter 4]

17 Jun 2024

dear respected editorials we have adjust all necessary comments please procced it for publication

---

## [Editor Report · Decision Letter 5]

21 Jun 2024

Effects of dry bio-slurry and nitrogen fertilizer on potato and wheat yields under rotation cropping system

PONE-D-23-10312R5

Dear Dr. Musse,

We’re pleased to inform you that your manuscript has been judged scientifically suitable for publication and will be formally accepted for publication once it meets all outstanding technical requirements.

Kind regards,

Ravinder Kumar, Ph.D.

Academic Editor

PLOS ONE

---

## [Editor Report · Acceptance letter]

26 Jun 2024

PONE-D-23-10312R5 

PLOS ONE

Dear Dr. Addis, 

I'm pleased to inform you that your manuscript has been deemed suitable for publication in PLOS ONE. Congratulations! Your manuscript is now being handed over to our production team.

Kind regards, 

on behalf of

Dr. Ravinder Kumar 

Academic Editor

PLOS ONE